# Characterization of a Spirit Beverage Produced with Strawberry Tree (*Arbutus unedo* L.) Fruit and Aged with Oak Wood at Laboratorial Scale

Ofélia Anjos [1,2,3,*], Soraia Inês Pedro [2], Débora Caramelo [3], Andreia Semedo [4], Carlos A. L. Antunes [1], Sara Canas [5,6] and Ilda Caldeira [5,6]

[1] IPCB–Polytechnic Institute of Castelo Branco, 6001-909 Castelo Branco, Portugal; carlosalbertoantunescb@gmail.com

[2] CEF–Forest Research Centre, School of Agriculture, University of Lisbon, 1349-017 Lisboa, Portugal; soraia_p1@hotmail.com

[3] Spectroscopy and Chromatography Laboratory, Centre of Plant Biotechnology of Beira Interior, 6001-909 Castelo Branco, Portugal; dbrcaramelo@gmail.com

[4] Faculty of Science and Technology (FCT), University of Cabo Verde (Uni-CV), Praia CP 379C, Cape Verde; andreiam.semedo@student.unicv.edu.cv

[5] INIAV–Dois Portos, Quinta da Almoínha, 2565-191 Dois Portos, Portugal; sara.canas@iniav.pt (S.C.); ilda.caldeira@iniav.pt (I.C.)

[6] MED–Mediterranean Institute for Agriculture, Environment and Development, University of Évora, Pólo da Mitra, Ap. 94, 7006-554 Évora, Portugal

* Correspondence: ofelia@ipcb.pt

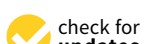



**Featured Application: This study will allow the producers of *Arbutus unedo* spirit to improve their products' diversity and offer aged Arbutus unedo spirit with the required analytical and sensory characteristics.**

**Abstract:** *Arbutus unedo* spirit is a valuable product in Mediterranean countries. This spirit is usually marketed in Portugal without wood ageing. This work aims to characterize the ageing effect on the *Arbutus unedo* spirit, for three and six months with oak wood (*Quercus robur* L.) submitted to different toasting levels, based on its chemical composition and its sensory properties. For this purpose, several parameters were analysed: acidity, pH, dry extract, and volatile compounds (methanol, acetaldehyde, ethyl acetate and fusel alcohols). The volatile compounds were identified by GC-MS and quantified by GC-FID. Sensory analysis was performed by a trained panel, who have profiled this beverage, as well as the changes acquired during ageing. Spectroscopic techniques, namely FTIR–ATR, were applied to discriminate the different beverages produced. The results highlighted an increase in *Arbutus unedo* spirit's quality with the wood contact, mainly based on the sensory attributes. Additionally, they showed that the best beverages were produced using oak wood with medium toasting levels during three months of ageing.

**Keywords:** *Arbutus unedo*; spirit; ageing; wood; volatile compounds; sensory analysis; FTIR-ATR

## 1. Introduction

*Arbutus unedo* L. (strawberry tree), growing in Portugal, originates from the Mediterranean basin. This species is also found in northeast Africa (except Egypt and Libya), western Asia, Canary Islands, and across southern Europe [1,2]. The fruit of the strawberry tree has different commercial uses, it can be consumed as fresh fruit, or used for ornamental applications, or for pharmaceutical and chemical industrial applications [3,4].

The fruits of the strawberry tree are usually harvested from wild plants or from orchards, and their fermentation process is traditionally made without crushing and without yeast inoculation. Thereafter, the distillation is performed by a discontinuous process, using copper alembics with separation of distillation fractions [5]. The obtained

distillate is usually marketed without ageing in wood, unlike other distillates such as whiskies or wine spirits.

In the last decade, some studies on this distillate have been carried out and the results have demonstrated that the fruit ripeness, the fermentation processes and the distillation technology are key factors in determining the final quality of this spirit drink [1,6,7].

The chemical composition of this alcoholic beverage must comply with European regulations [8]. In Portugal, an additional regulation is imposed on spirits with an alcohol strength higher than 42% [9] and there is a range or limit to the contents of other analytes, such as volatiles and copper [10,11], for the *Arbutus unedo* spirit (AUS).

Regarding the AUS sensory profile, few results have been published [12]. This spirit is colourless, with a clear and bright appearance, with odour notes of fruity and dried fruits, and a smooth and soft flavour [10,11]. The odour notes of the fruit used as raw material is common in different spirits, such as apple spirit [13] passion fruit spirit [14] grape marc spirit [15] and others [16].

Concerning the ageing of this beverage, to the best of our knowledge, only one study was carried out [17], evaluating the effect of oak wood and chestnut wood (using 50 L barrels). This study reported that chestnut wood imparted more colour and less taste than oak wood. According to this author, this technological option is preferable because it helps to maintain the flavour of this kind of distillate throughout the ageing time. The author also suggests an ageing time of less than 12 months in wooden barrels.

During the ageing period in wooden barrels, the spirits develop a characteristic flavour that increases the consumer's preferences [18]. It is known that the phenomena underlying ageing depends on the composition of the wine distillate, as well as on the characteristics of the wood, the cellar conditions, the technological operations carried out, and the duration of ageing.

Oak is the wood traditionally used in the ageing of wine spirits, especially wine originating from the French region of Limousin (mainly of *Quercus robur* L species). However, some research has been performed concerning the use of other oak species or other wood species in the aging of spirits [18–22].

The ageing phase in wooden barrels is part of the traditional production process of wine spirit and aims to add value to the product, which is already differentiated from other spirits. During this stage, several physical and chemical phenomena occur, such as the extraction of wood compounds, multiple reactions between these compounds and those of the wine distillate, and the evaporation of alcohol and water. Throughout the ageing process, the spirit acquires characteristics non-existent in the wine distillate, which contribute to an increase in overall quality. The most evident change occurs in the colour related to the wood-distillate interaction [23]. On the other hand, some features, such as the kinetics of odorant compounds and the chemical composition of the wood provided to spirit drinks their final and differentiate quality [24,25].

Recently, alternative ageing technologies have been studied for some spirits, namely the use of oak wood fragments, and the results suggest that they can be used for good cellar practice since they promote the acceleration of the ageing process [26,27].

The purpose of this pioneer work was to study an alternative process for the ageing of AUS, using Limousin oak wood staves with different toasting levels during three and six months, and to assess the differences promoted in the chemical composition and sensory profile of this spirit beverage.

## 2. Materials and Methods

### 2.1. Experimental Design and Sampling

A total of 12 samples of the *Arbutus unedo* spirit (produced at laboratorial scale; 2 L glass bottles) aged with Limousin oak (*Quercus robur* L.) for three and six months was analysed (Figure 1), covering the following groups:

L3–aged with light toasting wood for three months
M3–aged with medium toasting wood for three months

MP3–aged with medium plus toasting wood for three months
L6–aged with light toasting wood for six months
M6–aged with medium toasting wood for six months
MP6–aged with medium plus toasting wood for six months

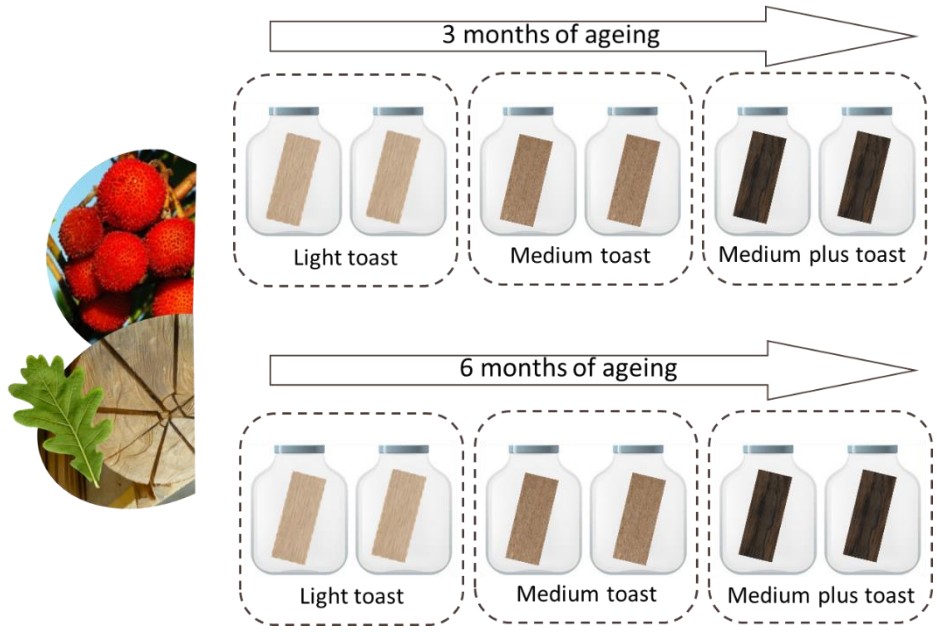

**Figure 1.** Scheme of the essay.

A sample of the *Arbutus unedo* spirit without wood ageing was used as a control.

The staves were purchased from J.M. Gonçalves cooperage (Palaçoulo, Portugal) and manufactured according to procedure and protected by intellectual property, using an industrial oven under controlled and differentiated temperature to attain the different toasting levels. The quantity of wood staves used in the essay was calculated according to the surface area to volume ratio of a 250 L barrel (85 cm$^2$/L), which is the dimension generally used in the aging of spirits.

For each modality, two essay replicates were made in glass bottles of 2 L, which were kept in the dark in a cold place. All analytical and sensory assessments were performed after one month of the ageing procedure

The ageing time established for this essay was only six months, as it was intended to increase the quality of the final product, but at the same time preserve the typical aroma of strawberry fruits in this spirit drink.

### 2.2. Analytical Procedures

#### 2.2.1. Physicochemical determinations

In order to characterise the different AUS samples, some analytical determinations were performed: alcoholic strength, density, dry extract, total acidity, fixed acidity, volatile acidity, pH, methanol, acetaldehyde, and higher alcohol and ethyl acetate concentration. All analyses were carried out in duplicate.

Alcohol strength was determined by distillation and electronic densimetry [28], using an electronic densimeter Model 5000 DMA brand Anton Paar (Graz, Austria). The results are presented as the volumetric percentage of ethanol in the beverage.

The dry extract was evaluated by the method proposed for wine spirits [28], which consists of weighing the residue after evaporation, at 100 °C, of the spirits. The results are expressed in mg/L.

Total acidity was evaluated by colorimetric titration, fixed acidity was evaluated by colorimetric titration of the water solution of dry extract and volatile acidity was

determined by calculation (total acidity minus fixed acidity) [28]. The results are expressed in grams of acetic acid per litre.

The pH was determined using a potentiometer (C3060, Consort- multi-parameter analyser, Belgium), according to the OIV method [28].

### 2.2.2. Volatile Composition

Methanol, acetaldehyde, ethyl acetate and higher alcohols of each AUS sample were quantified by gas chromatography-flame ionization detection (GC-FID). The compound's concentrations were determined by direct injection of the distillate resulting from the alcohol strength determination. Prior to injection, 1 mL of internal standard solution (4-methylpentan-2-ol) was added to 10 mL of each sample).

The GC-FID analysis was carried out using a Focus GC gas chromatograph (ThermoFinnigan, Milan, Italy), equipped with a flame ionization detector-FID (250 °C) and a fused silica capillary column of polyethylene glycol (DB-WAX, JW Scientific, Folsom, CA, USA), 60 m length, 0.32 mm i.d., 0.25 m film thickness. The samples were loaded (~1 μL) into the injector (200 °C) in split mode (split ratio 1:6) and the hydrogen flow (carrier gas) was 3.40 mL/min. The initial oven temperature was held at 35 °C for 8 min, then increased to 200 °C at 10 °C/min rate, which was kept for 1 min. Quantification was performed using hydroalcoholic solutions of standard compounds and analysed under the same chromatographic conditions.

The compounds' identification was carried out using GC-MS equipment (Magnum, Finnigan Mat, San Jose, CA, USA), under similar conditions regarding the chromatographic column, oven temperature and injector. The transfer line was kept at 250 °C, the carrier gas was helium (inlet pressure 83 KPa and split ratio 1:60), and the mass spectra were obtained in the electron impact (EI) mode (ionization energy, 70 eV) in full-scan mode (mass range m/z 20–340). The identification was made by comparing their mass with those of the NIST library and by analysing the mass spectra of standards.

### 2.2.3. Vibrational Spectroscopy

The spectra of AUS samples were obtained using the methodology described by Anjos [29]. AUS spectra were acquired with a FTIR-ATR (Fourier transform infrared spectroscopic method with platinum Attenuated Total Reflectance crystal) with a Bruker spectrometer (Alpha, Bruker Optic GmbH, Ettlingen, Germany). Five spectra per sample were obtained with 128 scans per spectrum at a spectral resolution of 8 $cm^{-1}$ in the range of 4000 to 450 $cm^{-1}$. The FTIR-ATR used was equipped with a flow-through cell with controlled temperature. The background was measured with distilled water and the cell was cleaned by the injection of water in the flow-through cell. The spectra were collected at constant room temperature (22 °C).

### 2.3. Semsory Analysis

The AUS samples were evaluated by a group of ten trained tasters according to the procedure previously described [30]. Two groups of sensory attributes were evaluated: (1) orthonasal aroma attributes—alcohol, fruity, vanilla, wood, rancid, spicy, caramel, toasted, dried fruits, smoke, coffee, sweet, green, tails, glue and caoutchouc; (2) gustatory attributes—sweetness, smooth, burning, astringency, roughness, bitter, body, unctuous, flavour complexity, flavour evolution, retronasal aroma and persistence.

All samples were adjusted, with water, to 40% *v/v*, and were kept in the dark at 20 °C until analysis.

The tasting session was carried out in the tasting room with individual white boots of Instituto Nacional de Investigação Agrária e Veterinária (INIAV). The sensory evaluation was conducted in the morning at 11:00 a.m. and 30 mL of each sample was assessed in standard wine-tasting glasses (ISO 3591) [31]. The samples were coded with three random digits and presented in balanced order to eliminate first-order carryover effects [32].

A structured scale was used to evaluate the sensory attributes (0—no perception to 5—highest perception) and to rate the overall quality of the AUS (0—lower quality to 20—highest quality).

*2.4. Data Analysis*

The results of different analytical determinations obtained for AUS samples were submitted to a post hoc test in a one-way analysis of variance in two steps. In the first step, the control was compared to the other modalities to identify the significant differences; in the second step, the ageing modalities were compared. Scheffe's test was applied for mean comparison.

The results were also examined through multivariate analysis, namely principal component analysis (PCA), to compare the similarity between samples.

All the calculations were performed using statistics from Statsoft (vs. 7.09, Statsoft Inc., Tulsa, OK, USA).

PCA was also performed in FTIR-ATR spectra of AUS to distinguish samples. Six replicated spectra for each beverage sample were collected and the analysis was performed with the average spectra of these measurements.

Spectral data analysis was carried out using UnscramblerX 10.5 (CAMO, Oslo, Norway).

## 3. Results and Discussion

*3.1. Physicochemical Analysis*

Table 1 summarises the results of the basic analytical determinations over the ageing time.

**Table 1.** ANOVA results and mean values of acidity, pH, and dry extract of the AUS aged with oak.

| Samples | Total Acidity | Fixed Acidity | Volatile Acidity | pH | Dry Extract (mg/L) |
|---|---|---|---|---|---|
| | (g of Acetic Acid/L) | | | | |
| C | 0.140 ± 0.00 * | 0.009 ± 0.001 * | 0.130 ± 0.001 * | 3.91 ± 0.02 | 10 ± 2.00 * |
| LT3 | 0.198 ± 0.030 a | 0.063 ± 0.010 a | 0.135 ± 0.024 a | 4.01 ± 0.09 a | 435 ± 65.7 a |
| MT3 | 0.210 ± 0.036 a | 0.075 ± 0.013 ab | 0.135 ± 0.044 a | 3.86 ± 0.01 a | 740 ± 24.5 d |
| MPT3 | 0.298 ± 0.025 bc | 0.070 ± 0.014 ab | 0.228 ± 0.029 b | 3.91 ± 0.02 a | 626 ± 11.8 c |
| LT6 | 0.265 ± 0.042 b | 0.078 ± 0.005 ab | 0.188 ± 0.039 ab | 3.77 ± 0.14 a | 534 ± 14 b |
| MT6 | 0.330 ± 0.024 c | 0.105 ± 0.006 c | 0.225 ± 0.026 b | 3.85 ± 0.03 a | 793 ± 53.8 d |
| MPT6 | 0.325 ± 0.031 c | 0.083 ± 0.005 b | 0.243 ± 0.030 c | 3.97 ± 0.03 a | 678 ± 59.1 c |

* Control denotes significant differences from the ageing modalities. Means within the same column followed by different letters are significantly different ($p < 0.05$) according to Scheffe's test.

The ANOVA results for acidity (Table 1) of the AUS reveal that the control spirit had significantly lower values compared to the aged ones. Additionally, the AUSs aged for 6 months presented significantly higher values than the spirits aged for 3 months. Concerning the toasting level, a trend to higher values of acidity in the spirits aged with medium toast level was observed.

The increase of acidity with ageing time and with toasting level intensity is also reported for other spirits [33–35].

pH values of a spirit drink usually vary with the ageing time, however, for these samples they were not significantly different over time. This result can be explained by the short ageing time set (three and six months), compared to what is normally used for other spirit drinks. The values of total acidity and pH (0.17–0.59 and 3.50–4.01, respectively) are within the range of values found in AUS in previous works [6,12,36].

The dry extract increased with the ageing time, but more significantly with the toasting level (Table 1), which is ascribed to the extent of changes caused by the toasting level on the chemical composition and on the structure of the wood. Results show that MT promoted more favourable conditions than LT, giving rise to higher extraction of compounds from the wood to the distillate, positively influencing the dry extract in both sampling times. However, MPT seemed to cause an imbalance between compound formation and degradation under the influence of a more intense toasting, resulting in lower dry extract in the

corresponding AUS samples than those of MT in both sampling times. These results are in accordance with those obtained in the ageing of other spirits, such as wine spirits [33] and plum distillates [27]. The control spirit presented a quit null dry extract denoting a good distillation process.

The values shown in Table 2 for alcohol strength were above the minimum required by the European regulation [8] and are within the range of values observed in other studies [6,12,36]. Alcohol strength of aged AUS present values ranging between 46.8% and 47.9% and shows an insignificant variation because this essay was made in bottles in which ethanol evaporation did not occur during the ageing process. Despite the small difference between the alcohol content of the control sample and the aged spirit samples, it proved to be significant in the statistical analysis. The explanation for this increase in alcohol content during ageing may be the occurrence of greater water sorption, as already verified for volatile compounds [37] or a faster penetration of water than ethanol into the wood, which greatly influences the impregnation kinetics in the beginning of ageing [38].

**Table 2.** Mean and Standard deviation of alcohol strength, methanol, acetaldehyde, and ethyl acetate of the AUS aged with oak wood.

| Code | Alcohol Strength (% *v/v*) | Methanol (mg/L) | Acetaldehyde (mg/L) | Ethyl Acetate (mg/L) |
|------|----------------------------|------------------|---------------------|----------------------|
| C    | 46.2 ± 0.00 *              | 4370 ± 8.54 *    | 115.4 ± 0.49 *      | 259.4 ± 2.12 *       |
| LT3  | 47.9 ± 0.11 b              | 4247.4 ± 83.60 a,b | 127.6 ± 2.31 b    | 245.1 ± 46.6 a       |
| MT3  | 46.9 ± 0.01 a              | 4225.0 ± 38.76 a | 124.7 ± 5.95 b      | 258.4 ± 9.70 a       |
| MPT3 | 46.9 ± 0.01 a              | 4225.4 ± 56.30 a | 122.4 ± 5.79 b      | 270.0 ± 7.41 a       |
| LT6  | 46.8 ± 0.13 a              | 4239.9 ± 37.54 b | 124.0 ± 4.17 b      | 266.6 ± 5.98 a       |
| MT6  | 47.0 ± 0.03 a              | 4294.9 ± 34.91 a,b | 123.5 ± 2.57 b    | 269.5 ± 3.49 a       |
| MPT6 | 47.0 ± 0.03 ab             | 4301.1 ± 63.75 b | 110.2 ± 6.52 a      | 266.7 ± 22.7 a       |

* Control denotes significant differences from the other modalities. Means within the same column followed by different letters are significantly different ($p < 0.05$) according to Scheffe's test.

Regarding the methanol content (Table 2), the significantly higher content in AUS aged after six months (between 4239.9 mg/L and 4341.1 mg/L) than after 3 months (between 4225.0 mg/L to 4239.9 mg/L) should be stressed.

Methanol is an undesirable compound in all spirits, given its toxicity [39], and its restrictive control is mandatory, but it does not directly affect the flavour of the distillate. All the values found in this work were lower than the legal limit.

These results are not in accordance with some authors, who reported a tendency for low amounts of methanol as the ageing time progressed. Such a discrepancy can be assigned to the methanol oxidation and formation of diethoxymethane as consequence of acetalization reaction [26,40,41]. Moreover, the cited studies were based on quite a long ageing period (2 to 5 years), which was enough for the aforementioned chemical changes. In our study, based on a six-month period at laboratorial scale, the methanol content in the AUS with six months of ageing was significantly higher than in the other ones, but only by a small percentage, which could be indicative of the wood variability and the usual variability observed in the first stages of the ageing process [19].

Higher alcohols (Table 3) are important compounds related to the quality and aroma of spirit drinks and are considered as part of the aromatic skeleton of fruit distillates [42]. The values of control AUS are similar to those previously reported by other authors [6,12,36]. Concerning the butan-1-ol, propan-1-ol, 2-methylpropan-1-ol and butan-2-ol, no significant differences were found between samples. Only for isoamyl alcohols (2+3-methylbutan-1-ol) the values of the aged beverages were lower than those observed in the control. Concerning the aged wood samples, isoamyl alcohols, have significantly lower concentration in the beverages aged with MPT wood. The effect of the ageing time was not significant for the concentration of 2+3-methylbutan-1-ol.

**Table 3.** Mean and Standard deviation of higher alcohol of the AUS aged with oak wood.

| Code | butan-1-ol (mg/L) | propan-1-ol (mg/L) | 2-methylpropan-1-ol (mg/L) | butan-2-ol (mg/L) | 2+3-methylbutan-1-ol (mg/L) |
|------|-------------------|--------------------|-----------------------------|-------------------|------------------------------|
| C | 1.79 ± 0.02 | 84.88 ± 0.05 | 200.57 ± 0.34 | 1.66 ± 0.11 | 839.77 ± 1.34 * |
| LT3 | 1.85 ± 0.31 a | 83.04 ± 1.57 a | 197.53 ± 1.56 a | 1.82 ± 0.21 a | 825.07 ± 4.98 b |
| MT3 | 1.96 ± 0.06 a | 82.87 ± 0.49 a | 197.18 ± 1.66 a | 1.71 ± 0.27 a | 824.44 ± 6.22 b |
| MPT3 | 1.69 ± 0.30 a | 82.35 ± 1.22 a | 194.46 ± 2.77 a | 1.86 ± 0.06 a | 808.29 ± 12.35 a |
| LT6 | 1.90 ± 0.14 a | 83.39 ± 0.28 a | 196.00 ± 1.21 a | 2.17 ± 0.02 a | 819.12 ± 1.79 b |
| MT6 | 1.87 ± 0.11 a | 83.59 ± 0.87 a | 196.45 ± 1.53 a | 2.08 ± 0.21 a | 822.22 ± 8.05 b |
| MPT6 | 1.86 ± 0.21 a | 83.61 ± 0.59 a | 197.79 ± 2.21 a | 1.70 ± 0.31 a | 810.18 ± 7.94 a |

* Control denotes significant differences from the other modalities. Means within the same column followed by different letters are significantly different ($p < 0.05$).

Rodríguez Madrera et al. [43] identified evaporation, oxidation, esterification, and sorption as the main sources to explain the decrease in higher alcohols during ageing. Sorption phenomenon is associated with wood species, porosity, and the toasting level [37], and may also explain the decrease in methanol and 2+3metyl-1-butanol contents in aged samples, in comparison to the control sample.

The PCA (standardized biplot of loadings and scores) made with the physicochemical data confirmed the above-mentioned differentiation between three and six months of ageing (Figure 2). The medium and medium plus toasting level were only separated by the PC2, which only explains 20% of the total variation. The two principal components explain 64.1% of the total variance, but the third component which only explains 15.8% of the variance (plot not shown) still separates the same aforementioned groups.

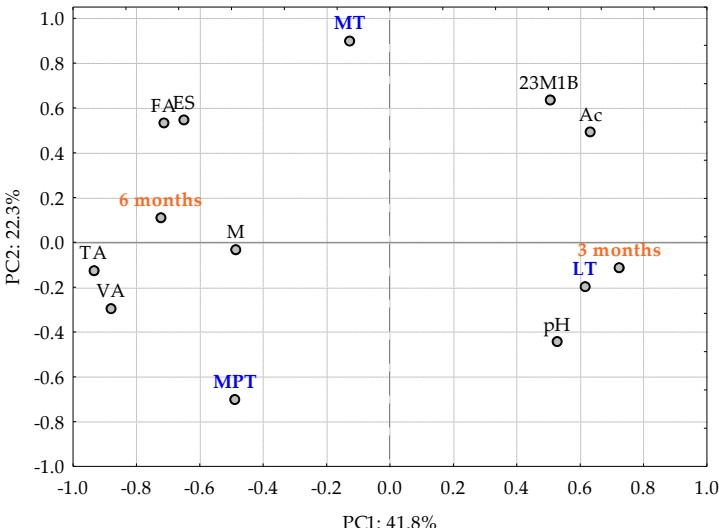

**Figure 2.** Principal component analysis of the physicochemical determination, and significant volatile compounds of *Arbutus unedo* spirits aged with oak, with three different toasting levels (LT–AUS aged in light toasting oak wood; MT–AUS aged in medium toasting oak wood; MPT–aged with medium plus toast oak wood) during three and six months. M—Methanol (g/mL); Ac—Acetaldehyde (mg/L); 23M1B—2+3-methylbutan-1-ol (mg/L); DS—dry extract (g/L); VA—Volatile acidity; TA–total acidity; FA–fixed acidity.

The samples aged for three months, and samples aged with light toasting levels are placed at the positive side of component 1, which is related to high values of pH, acetaldehyde and 2+3-methylbutan-1-ol. Samples aged for six months and samples with MPT and MT toasting levels are at the negative side of the PC1, which had a positive correlation with acidity, methanol content and dry extract. These results confirm and support the abovementioned observations.

### 3.2. Sensory Properties

The average results and the statistical results for AUS sensory assessment are shown in Table 4; the attributes which are significantly different among samples are highlighted. Complementarily, Figure 3 illustrates the sensory profile based on such significant attributes.

**Table 4.** Results of the sensory analysis of AUS regarding olfactory and gustatory attributes.

|  | Attributes | C | LT3 | MT3 | MPT3 | LT6 | MT6 | MP 6 |
|---|---|---|---|---|---|---|---|---|
| Olfactory | Alcohol | 2.8 | 2.8 | 2.3 | 2.6 | 2.6 | 2.6 | 2.8 |
|  | Fruity | 2.7 | 1.6 | 1.5 | 1.8 | 1.5 | 1.7 | 1.7 |
|  | Floral | 0.8 | 1.4 | 0.9 | 1.1 | 1.0 | 1.0 | 1.3 |
|  | **Vanilla** | 0.2 a | 1.2 b | 2.0 c | 1.0 b | 1.2 b | 1.6 bc | 1.8 c |
|  | **Woody** | 0.1 a | 1.9 ab | 1.9 b | 1.8 b | 1.1 ab | 1.9 b | 2.0 b |
|  | Rancid | 0.6 | 0.4 | 0.8 | 0.4 | 0.6 | 0.6 | 0.9 |
|  | Spicy | 0.8 | 1.3 | 1.6 | 1.4 | 1.3 | 1.5 | 1.5 |
|  | **Caramel** | 0.3 a | 0.7 ab | 1.7 d | 1.4 c | 0.8 ab | 1.1 bc | 1.5 c |
|  | **Toasted** | 0.5 a | 0.9 ab | 1.6 b | 1.3 b | 1.2 b | 1.3 b | 1.3 b |
|  | **Dried fruits** | 1.3 a | 1.4 ab | 1.8 b | 1.0 a | 1.2 a | 1.5 ab | 1.8 b |
|  | Smoke | 0.7 | 0.7 | 0.9 | 1.0 | 0.9 | 1.0 | 1.0 |
|  | **Coffee** | 0.1 a | 0.1 a | 0.3 ab | 0.3 ab | 0.4 ab | 0.3 ab | 0.7 b |
|  | **Sweet** | 1.5 b | 1.2 ab | 1.6 b | 0.8 a | 1.1 a | 1.1 a | 1.1 a |
|  | Green | 1.0 | 0.3 | 0.4 | 0.2 | 0.4 | 0.2 | 0.3 |
|  | Tails | 0.3 | 0.0 | 0.0 | 0.0 | 0.1 | 0.0 | 0.0 |
|  | Glue | 0.0 | 0.5 | 0.2 | 0.4 | 0.5 | 0.3 | 0.5 |
|  | Caoutchouc | 0.4 | 0.2 | 0.1 | 0.1 | 0.2 | 0.1 | 0.2 |
| Gustatory | Sweetness | 2.7 | 2.1 | 2.6 | 2.2 | 2.4 | 2.2 | 2.3 |
|  | **Smooth** | 2.4 a | 2.5 a | 3.1 b | 2.7 ab | 3.1 b | 2.5 a | 2.8 ab |
|  | **Burning** | 3.2 b | 2.7 a | 2.5 a | 2.7 a | 2.6 a | 2.6 a | 2.5 a |
|  | Astringency | 1.7 | 1.6 | 1.5 | 1.4 | 1.5 | 1.7 | 1.6 |
|  | Roughness | 2.2 | 1.8 | 1.6 | 1.8 | 1.7 | 1.7 | 1.9 |
|  | Bitterness | 1.0 | 1.2 | 0.9 | 1.3 | 1 | 1.2 | 1.3 |
|  | **Body** | 2.5 a | 2.4 a | 3.0 b | 2.9 ab | 2.7 ab | 2.9 ab | 2.9 ab |
|  | Unctuous | 1.8 | 2.2 | 2.6 | 2.0 | 2.4 | 1.9 | 2.2 |
|  | **Flavour evolution** | 1.9 a | 2.4 ab | 2.8 b | 2.8 b | 2.7 b | 2.6 b | 2.8 b |
|  | **Flavour complexity** | 2.8 a | 2.8 a | 3.2 ab | 3.0 ab | 3.3 b | 3.0 ab | 3.0 ab |
|  | **Retronasal aroma** | 2.9 a | 2.7 a | 3.3 b | 3.1 ab | 3.1 ab | 3.0 ab | 3.0 ab |
| **Aroma quality** |  | 12.8 a | 13.9 b | 15.2 c | 14.0 b | 14.1 b | 14.2 b | 14.3 b |
| **Flavour quality** |  | 13.2 a | 13.6 ab | 15.0 c | 14.2 bc | 13.6 ab | 14.4 bc | 14.2 bc |
| **Overall quality** |  | 13.2 a | 13.7 b | 15.0 c | 14.1 b | 14.2 b | 14.3 b | 14.1 b |

The variance analysis showed a marked effect of ageing on the following sensory attributes of the AUSs: (1) olfactory attributes: vanilla, woody, caramel, toasted, dried fruits, coffee, sweet; (2) gustatory attributes: smooth, burning, body, flavour evolution, flavour complexity, retronasal aroma; (3) overall quality including aroma and flavour (Table 4). The sensory attributes without influence from ageing are greenish; alcohol; fruity; floral; rancid; spicy; smoke; green; tails; glue; caoutchouc; sweetness; astringency; roughness; bitterness; unctuous. As far as we know, no studies were performed on the sensory characterization of the AUS, and more studies are needed to define the sensory fingerprint of this beverage.

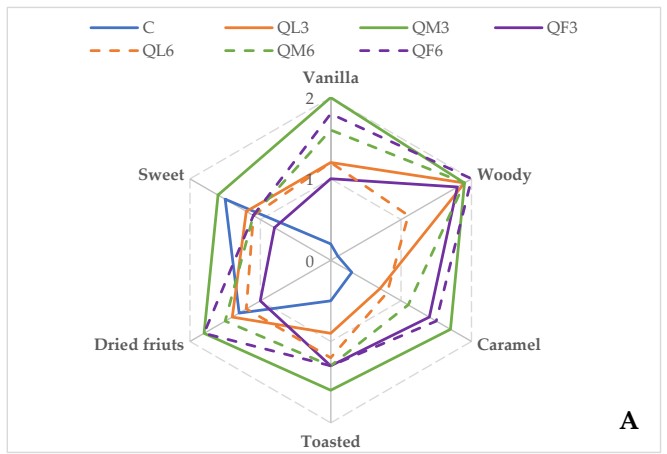

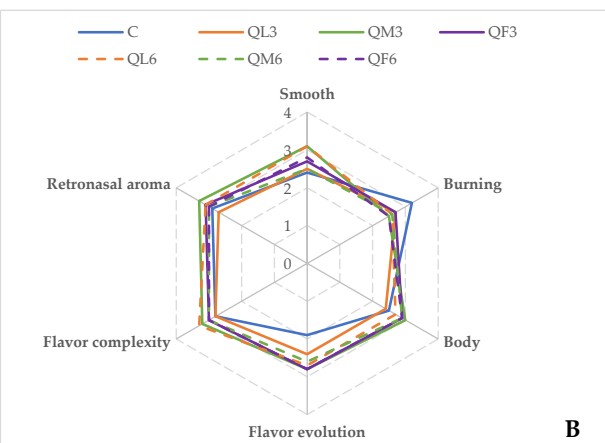

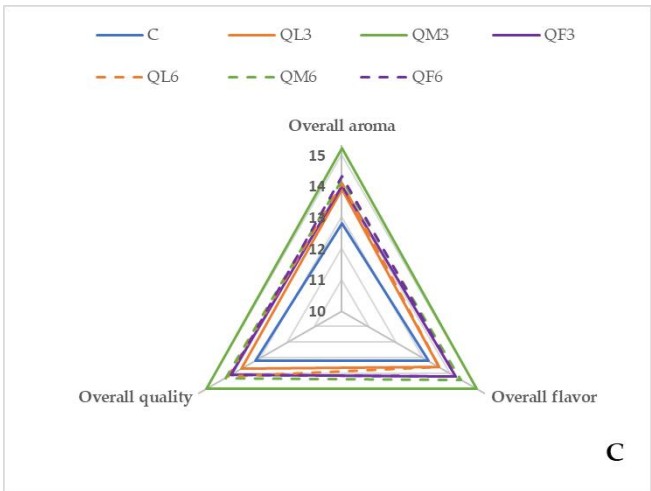

**Figure 3.** Olfactory (**A**), gustatory (**B**) and overall quality (**C**) profiles based on the average panel attributes scores of the *AUS*.

Regarding the olfactory attributes, ANOVA showed higher discrimination according to the toasting level than to the ageing time for the different spirits analysed. In this group, vanilla, woody, caramel, toasted, dried fruits, coffee, and sweet presented significant differences. Comparing these attributes, the more balanced AUS were those aged with MT, which also exhibited lower variation between three and six months of ageing. However, the effect of the toasting level is more evident than the observed ageing time.

The intensities of vanilla, woody, caramel, toasted, dried fruits, smooth, body, flavour evolution, flavour complexity and retronasal aroma increased from the light toast level to the medium toasting level. The aged samples became less sweet and burning with the ageing process. Other authors also observed an increase in the attributes of vanilla, spicy, unctuous, flavour evolution, retronasal aroma and flavour complexity over the ageing time [22,44] and, although the ageing period in this study was short, our results are in accordance with this trend.

The overall quality also increased throughout the ageing process. The lowest values were associated with light toasting. The highest scores of overall quality, aroma quality and flavour quality were assigned to the AUS aged for three months with medium toasting levels. These results confirm that a short stage in wood, particularly with medium toasting levels, allows an increase in the AUS' quality.

PCA of sensory results was also carried out (Figure 4). PC1 explains 41.6% of the total variation, discriminating the AUS aged with wood of light toasting from the other toasting levels. This discrimination is similar to those attained for the physicochemical characteristics (Figure 3). The AUS aged with medium toasting presented higher values of

vanilla, woody, caramel, toasted, smooth, body, flavour evolution, retronasal aroma, aroma quality, flavour quality, and overall quality. On the other hand, AUS aged with medium toasting presented higher notes of coffee and woody. A higher intensity of burning was observed in the AUS aged with light toasting level wood. The second principal component explains 15.0% of the total variation and makes the separation between AUS aged with medium and medium plus toasting levels; dried fruits and sweetness attributes were the main loading vectors.

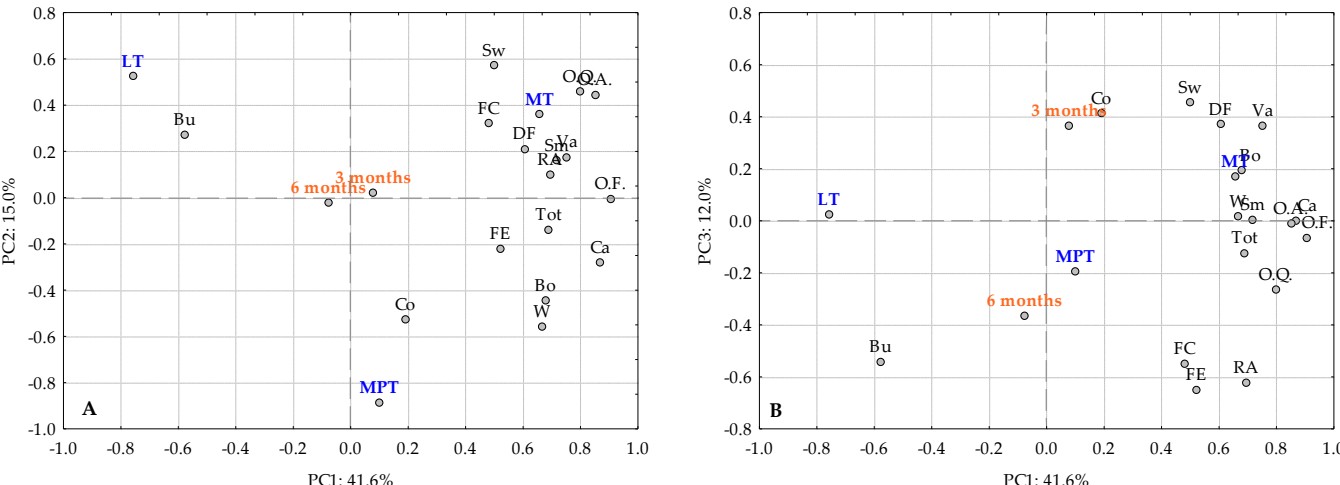

**Figure 4.** Principal component analysis of AUS aged with oak wood aged during three and six months for: (**A**)—PC1 vs. PC2; (**B**)—PC1 vs. PC3. LT—aged in light toast oak wood; MT—aged medium toast oak wood; MPT—aged with medium plus toast oak wood; Va—Vanilla; W—woody; Ca—caramel; Tot—toasted; DF—dried fruits; Co—coffee; Sw—sweet; Sm—smooth; Bu—burning; Bo—body; FE—flavour evolution; FC—flavour complexity; RA—retronasal aroma; O.A.—overall aroma; O.F.—overall flavour; O.Q.—overall quality.

PC3, which accounts for 12.0% of the total variation, explains the difference between samples aged for three months and six months; coffee, burning and flavour complexity and evolution; retronasal aroma and sweet were the main loading vectors.

### 3.3. FTIR-ATR Analysis

The collected absorbance spectra of all AUS samples studied are represented in Figure 5. The spectra profile are in accordance with those previously reported for wine spirits and grape marc spirits [29,45–49]. Since the representative absorbance spectra are very well explained in these works, only a summary is shown in the present one. Briefly, the AUS IR region has important information from 3000 to 2900 cm$^{-1}$ [O–H stretching, C–H stretching of $CH_3$ and $CH_2$], and from 1500 to 860 cm$^{-1}$ (C–C absorption bands, C–O vibrations, C–OH bending deformation, C-H bond stretching and C=O and C=C groups) [49–53].

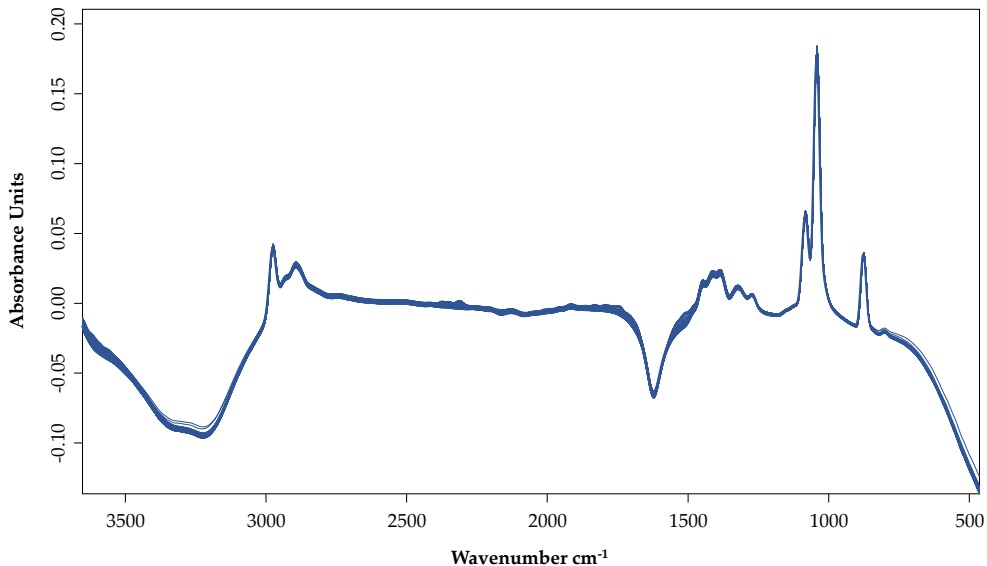

**Figure 5.** FTIR-ATR absorbance spectra of all AUS samples.

The regions aforementioned, identified as relevant for AUS discrimination, were analysed and the results were plotted in the PCA summarized in Figure 6.

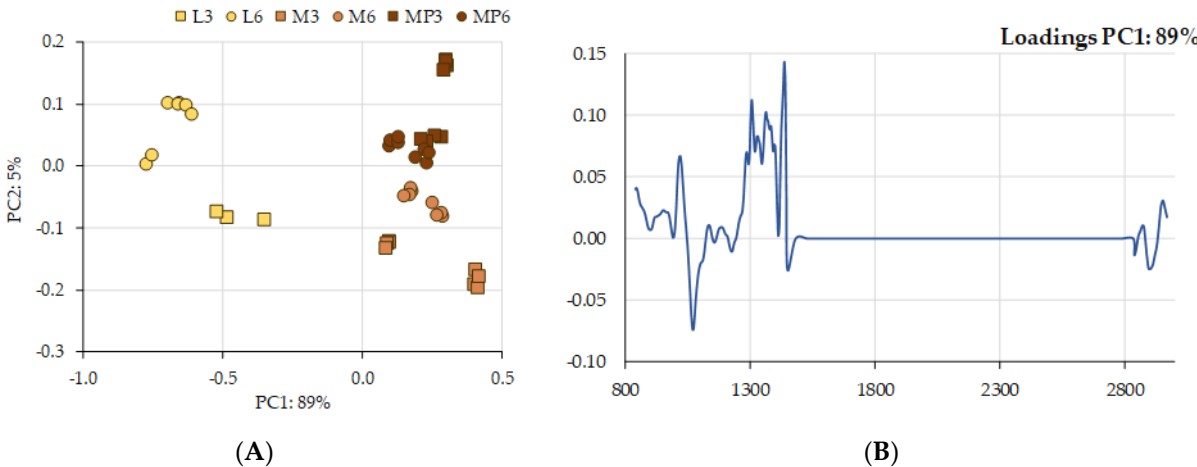

**(A)**                                                 **(B)**

**Figure 6.** Score plot of two principal components from spectral data acquired with FTIT-ATR of AUS, concerning the toasting level of the wood used in the ageing process for three and 6 months. (**A**)—Score plot, (**B**)—loading plot.

Figure 6A shows the score plot of two principal components made with FTIR-ATR spectra for AUS aged according to the wood toasting levels and the ageing time. The first two principal components accounted for 94% of the total variance, which is mainly explained by the toasting levels. In the first ANOVA performed (Table 1), the toasting level also had a significant effect on the physicochemical characteristics and on some volatile compounds. This effect was already noticed for wine spirit aged at an industrial scale by other authors [54,55]. They reported an increasing trend of some volatile and low molecular weight compounds over the ageing time. However, it is important to emphasize that the present study was made intentionally in a short period to preserve the aromatic profile of the strawberry tree's fruits in the beverage.

The separation between MT and MPT is also notified but only by the PC2, which explains 5% of the total variation.

The most relevant region to explain the observed variation was between 1300 cm$^{-1}$ and 1500 cm$^{-1}$, and from 2850 cm$^{-1}$ to 3000 cm$^{-1}$. Shurvell (2001) [53] reported that the

peaks at 1085 cm$^{-1}$ and 1043 cm$^{-1}$ are important for ethanol and methanol quantification, corresponding to the C–O stretch absorption bands. Despite the high effect of the toasting level, it was also possible to identify the effect of the ageing time, which confirmed the observed differences and the possible applicability of FTIR-ATR to distinguish these kind of samples.

These results showed that FTIR-ATR is a useful technique to discriminate AUS aged with wood according to the toasting level, and also to the ageing time. The applicability of the FTIR-ATR technique was also validated to perform calibration models of some volatile compounds for this alcoholic beverage [46,47,56–58].

## 4. Conclusions

The effect of alternative ageing with oak wood staves on the AUS was analysed for the first time. The results confirmed the positive influence of wood contact on the volatile and sensory profile of AUS. Among higher alcohols, the significant influence of wood ageing was only observed for the 2+3-methylbutan-1-ol.

Concerning the sensory overall quality of the aged AUS, the best results were obtained with the medium toasting level for three months of ageing.

This ageing technology allowed for an increase in the beverages' overall quality and, at the same time, preserved the natural aromas of AUS. In addition, the ageing of AUS contributed to a decrease in the methanol content, with subsequent advantages.

It is possible to conclude that the FTIR-ATR technique allowed discriminating aged AUS based on the ageing time and toasting level, as well identifying the more promising regions to perform calibration models of some volatile compounds for this alcoholic beverage.

**Author Contributions:** Conceptualization. O.A. and I.C.; methodology. O.A. and I.C.; formal analysis. S.I.P., A.S., D.C., C.A.L.A.; investigation. all; resources. O.A., S.C. and I.C.; writing—original draft preparation. O.A. and D.C.; review. all; editing. O.A., S.C. and I.C.; project administration. O.A. and I.C.; funding acquisition. O.A., S.C. and I.C. All authors have read and agreed to the published version of the manuscript.

**Funding:** This research received no external funding.

**Institutional Review Board Statement:** Not applicable.

**Informed Consent Statement:** Not applicable.

**Data Availability Statement:** Data is contained within the article.

**Acknowledgments:** This research was funded by the Forest Research Centre, a research unit funded by Fundação para a Ciência e a Tecnologia I.P. (FCT), Portugal (UIDB/00239/2020) and Mediterranean Institute for Agriculture, Environment and Development a research unit funded by Fundação para a Ciência e a Tecnologia I.P. (FCT), Portugal UIDB/05183/2020. The authors thank Mendes & Mendes, LDA distillery for the Arbutus unedo distillate supplied. The authors are grateful to Deolinda Mota from INIAV—Pólo de Dois Portos for their technical support in GC-FID analysis. The authors also thank the tasters of INIAV sensory panel for their persistence and availability.

**Conflicts of Interest:** The authors declare no conflict of interest.

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
