# Peer review of "Characterization of a Spirit Beverage Produced with Strawberry Tree (Arbutus unedo L.) Fruit and Aged with Oak Wood at Laboratorial Scale"

_applsci, doi:10.3390/app11115065_

Round 1
Reviewer 1 Report
The article describes the effect of wood ageing in Arbutus unedo spirits (AUS). This is an alcoholic beverage commonly commercialized without wood ageing process. The research is focused on the ageing of AUS in differently toasted oaks for 3 and 6 months and related results. The authors purpose an overall approach combining sensory profilation and a general set of analysis to describe the effect of ageing on these beverages. A correct statistical approach completes the paper and support the purposed results. The article is well written and clear but there are many points to clarify and several analyses are missing.
ABSTRACT: In the abstract (line 27) the authors say that volatile compounds were identified with GC-MS and quantified with GC-FID but in all the other section there are no words about this untargeted approach.
INTRODUCTION: Too many self-citations (up to 14 of 53, a 25%). I understand that AUS are not deeply studied like other beverages but the authors are invited to cite also other papers.
MATERIAL AND METHODS:
- The authors have to quantifiy “Low”, “Medium” and “Medium Plus” toasting levels and to describe the toasting procedure.
- Line 100 typing error: used n (in?) the spirits ageing
- What were the quality controls in the volatile analysis? Do the authors used Initial and Continuous Calibration Verifications? Many details about the analytical process (LOD, LOD, linearity, validation) are missing. Please complete this part in order to allow other scientists to refer to it.
- How much time passed from the end of the ageing process and the beginning of the analysis?
- In line 216 the authors declare that the control sample has a significantly lower concentration of alcohol strength compared to aged samples. Comparing the control value (46.2%) with aged samples the spread is usually within a 1% of difference so it does not seem to be that much different.
RESULTS AND DISCUSSION:
- In line 231-233 are the authors able to demonstrate the occurrence of that reaction? Was the concentration of diethoxymethane been measured? Since in line 27 the presence of a GC-MS in the lab was mentioned it would be interesting to use it to measure diethoxymethane.
- In the “sensory properties” section the authors describe many interesting nuances and sensory attributes due to the ageing process (page 8); have the related molecules been analyzed using an instrumental technique? An instrumental approach to these key analytes is mandatory.
My overall rating about this article is that it could be interesting but more instrumental analyses are required to strengthen conclusions and to clarify many points described by sensory profiling. As suggested by the authors only a paper [11] was focused on the oak ageing of AUS but in that case many GC-MS and HPLC complete the analytical pattern with instrumental data. No data regarding carbonyls, esters, vanilline and many other ageing-markers are reported. From my point of view the paper must be completed with more targeted instrumental analyses.
Author Response
We thank the comments and suggestions posed by the Reviewer 1. All of them were taken into account to increase the manuscript’s quality. Responses to the Reviewer comments and references to changes made in the manuscript are presented below.
Accordingly, the manuscript was carefully reviewed, and the changes made are marked with track changes.

Reviewer 2 Report
This work “Characterization of a spirit beverage produced with strawberry tree (Arbutus unedo L.) fruit and aged with oak wood at laboratorial scale.” aims to study an alternative process for the ageing of AUS using Limousin oak wood staves with different toasting levels, and to assess the differences promoted in the chemical composition and sensory profile of this spirit beverage.
The authors already published a paper within this subject:
Anjos, O.; Canas, S.; Gonçalves, J.C.; Caldeira, I. Development of a Spirit Drink Produced with Strawberry Tree (Arbutus unedo L.) Fruit and Honey. Beverages 2020, 6, 38. https://doi.org/10.3390/beverages6020038
The topic in question is relevant and has practical application in the spirits industry.
The authors addressed well the subject in question in the introduction, proper experiments were made, and the data obtained are relevant.
I only have a few comments that should be addressed before publication:
Line 19 and 20 - Arbutus unedo should be in italic (Arbutus unedo)
Line 132 – sample.)., remove one point.
Figure 3 C – Overall falvor, should be Overall flavor.
Line 326 - PC1 vs PC2. it should be PC1 vs PC3.
Line 372 - three months is in italic…I think there is no need for it.
So, in my opinion, the article needs a minor revision before publishing.
Author Response
We thank the comments and suggestions posed by the Reviewer 2. All of them were taken into account to increase the manuscript’s quality. Responses to the Reviewer 2 comments and references to changes made in the manuscript are presented below.
Accordingly, the manuscript was carefully reviewed, and the changes made are marked with track changes.

Reviewer 3 Report
The work presents high scientific potential. Introduction part is sufficient. Analytical techniques have been chosen accurately.
Only: Line 89: produced at laboratorial scale) – detailed information (in liters) would be more clear for readers.
Statistical analysis of data is also well chosen, helpful to organize the results obtained. Tables and Figures are clear and communicative. Discussion of the obtained results OK. Only conclusions are too poor in my opinion.
As an example: Shorter ageing can be an alternative to a classic, because 1) modifies the aroma of wine and at the same time allows the preservation of natural aromas of wine and 2) limiting the concentration of methanol.
Or any conclusion from the last paragraph (“These results showed that FTIR-ATRis a useful tecnique to discriminateAUS aged with wood according to thetoasting level,and also to the ageing time. The aplicability of FTIR-ATR technique was also validated to perform calibration models of some volatile compounds for this alcoholic beverages”).
Author Response
We thank the comments and suggestions posed by the Reviewer 3. All of them were taken into account to increase the manuscript’s quality. Responses to the Reviewer 3 comments and references to changes made in the manuscript are presented below.
Accordingly, the manuscript was carefully reviewed, and the changes made are marked with track changes.

Round 2
Reviewer 1 Report
The authors modified the article in some areas, adding useful details and improving the overall quality. Thanks to this revision, the article is more robust, the deductions are better supported and the analytical approach chosen by the authors is also clearer. Reading is fluid and the bibliography has been enriched by adding other sources. The authors have placed themselves in an extremely constructive way and have interpreted my suggestions in the best possible way. I appreciated the changes made and I find that the revised article can be published.